# Effects of General Physical Activity Promoting Interventions on Functional Outcomes in Patients Hospitalized over 48 Hours: A Systematic Review and Meta-Analysis of Randomized Controlled Trials

**DOI:** 10.3390/ijerph18031233

**Published:** 2021-01-29

**Authors:** Joost P. H. Seeger, Niek Koenders, J. Bart Staal, Thomas J. Hoogeboom

**Affiliations:** 1Research Group Musculoskeletal Rehabilitation, HAN University of Applied Sciences, 6825 EN Nijmegen, The Netherlands; Joost.Seeger@han.nl (J.P.H.S.); Bart.Staal@han.nl (J.B.S.); 2Department of Rehabilitation, Radboud Institute for Health Sciences, Radboud University Medical Center, 6525 GC Nijmegen, The Netherlands; 3IQ Healthcare, Radboud Institute for Health Sciences, Radboud University Medical Center, 6500 HB Nijmegen, The Netherlands; Thomas.Hoogeboom@radboudumc.nl

**Keywords:** physical activity, physical functioning, exercising, walking, mobilization, mobility, length of stay, hospitalization

## Abstract

Low physical activity of patients is a global problem and associated with loss of strength and independent mobility. This study analyzes the effect of general physical activity promoting interventions on functional and hospital outcomes in patients hospitalized over 48 h. Five electronic databases were searched for randomized controlled trials. For outcomes reported in two studies or more, a meta-analysis was performed to test between-group differences (intervention versus control) using a random-effects model. The Grading of Recommendations Assessment, Development, and Evaluation (GRADE) approach was used to rate the certainty of evidence for each outcome. Out of 23,302 identified studies, we included four studies (in total *n* = 368 participants). We found with moderate certainty of evidence 0 reported falls in the intervention (*n* = 126) versus five reported falls in the control (*n* = 122), a non-statistically significant difference between intervention and control groups (*p* = 0.06). In addition, we found with (very) low certainty of evidence no statistically significant differences between groups on activities of daily living (ADL-activity) and time spent standing and walking. Overall, we found no conclusive evidence on the effect of general physical activity promoting interventions on functional outcomes. More research is needed to understand and improve the effect of general physical activity promoting interventions for patients during the hospital stay.

## 1. Introduction

Sedentary behavior during the hospital stay is common in patients admitted to the hospital [1]. Several studies estimate that hospitalized patients who can walk spend approximately 70–82% of their time during daytime lying in bed [1,2,3]. Excessive bed rest can lead to functional decline and deconditioning [4,5], which may result in complications, increased hospital readmissions and health problems that are not directly related to the primary cause for hospitalization [6,7,8]. This phenomenon is better known as hospitalization-associated disability: an avoidable and unnecessary increasing functional decline in patients that occurs during care [9].

On the positive side, increasing physical activity in patients during their hospital stay shows beneficial effects on their health status [10]. Even small amounts of physical activity are known to reduce the risk of disease and disability in public health [11,12,13]. Increasing inpatient physical activity might, therefore, counteract the negative consequences of hospitalization-associated disability. Higher levels of physical activity are related to better functional outcomes [14], reduced length of hospital stay [15,16], and diminished readmission [17]. It is our hypothesis that the negative consequences of sedentary behavior reduce when inpatient physical activity levels increase with general physical activity promoting interventions. Therefore, the current study aims to estimate the effect of general physical activity promoting interventions on functional and hospital outcomes in patients hospitalized over 48 h.

## 2. Materials and Methods

This systematic review and meta-analysis of randomized controlled trials was conducted following the guidelines of the preferred reporting items for systematic reviews and meta-analyses (PRISMA) statement [18], and the Cochrane handbook [19]. This review was a priori registered in the International Prospective Register of Systematic Reviews (PROSPERO) database: CRD42017059178. We used the definition of physical activity as provided by Caspersen et al.: ‘physical activity is any bodily movement produced by skeletal muscles that requires energy expenditure’ [20]. We defined general physical activity promoting interventions as non-disease-specific interventions aiming to promote the physical activity of patients during their hospital stay, which could be administered to patients with different medical indications without the need for supervision of specialized staff (unsupervised). General physical activity promoting interventions should be applicable to a broad range of patients; interventions used in specific patient populations were considered specific interventions. In addition, general physical activity promoting interventions should not consist solely of supervised exercises, as these interventions were considered (physical) exercise therapy rather than general physical activity promoting interventions [21].

We searched five electronic databases for published studies up to January 2020: MEDLINE, Cochrane Central Register of Controlled Trials, EMBASE, CINAHL, and PEDro. The search comprised studies that included patients, hospitalized over 48 h. No selection was made based on the reason for hospital admission (condition), age or other patient characteristics. Types of interventions that were considered relevant were: physical activity promoting interventions, early ambulation, mobility programs, exercise, fitness, locomotion, stepping, and self-care. Ineligible interventions were supervised training programs, high-intensity interval training, and disease-specific exercise programs, because these interventions were not considered as generic but as specific physical activity promoting interventions. No selection was made on the minimum duration or intensity of physical activity stimulation. Outcomes of interest in the search were: activities of daily living, muscle strength, quality of life, functional recovery, functional impairment, functional decline, disability, and inability (functional outcomes); length of hospital stay, patient readmission, and patient discharge (hospital outcomes). The search was limited to studies published in the English language. The complete search strategies were constructed with the support of an experienced librarian (OYC), see Appendix A.

We included randomized controlled trials that compared usual hospital care with usual care and the addition of general physical activity promoting interventions in hospitalized patients for at least 48 h. Studies were eligible if the intervention was (1) studied according to a randomized controlled trial design; (2) non-disease-specific; (3) not individually tailored; (4) conducted in a hospital; (5) without specialized staff supervising the intervention (unsupervised); and (6) was evaluated on functional or hospital outcomes. Studies were excluded if the intervention was conducted as part of a (home-based) exercise program.

Titles and abstracts of retrieved studies were independently screened by two reviewers (JS and NK screened studies published until May 2018, and EK and RB screened studies published between June 2018 and January 2020). Studies that provided insufficient information in the abstract regarding the eligibility criteria were retrieved for full-text evaluation (NK). Two reviewers independently evaluated full-text studies and determined their eligibility for inclusion in our review. Disagreements were resolved by consensus and, if disagreement persisted, by another review author (TH).

Functional outcomes were used to assess the effect of physical activity promotion on (instrumental) activities of daily living. The studies included information on: reported falls, activities of daily living, time spent in different levels of physical activity (time spent lying, sitting, standing, cycling, or walking), community mobility, hospital mobility, and revolutions cycled [22]. In this review and meta-analysis, community mobility was defined as ‘the ability of a person to purposefully move out of the room in which the person sleeps to another area in a specific time period’ [23].

Two reviewers (JS and NK) used standardized forms to independently extract the following data from each eligible study: study characteristics such as authors, year of publication, setting, type of intervention, and follow-up duration; study population characteristics such as age and gender; and study outcomes. Details on the type of intervention and usual care were extracted, and, if available, information about the frequency, intensity, and time. Details on the intervention were extracted and reported using the Template for Intervention Description and Replication (TIDieR) checklist, see Appendix A [24].

Two reviewers (JS and NK) independently assessed the risk of bias in each included study using the Cochrane ‘Risk of bias-2’ tool [25,26]. Tables with the completed Risk of Bias-2 assessment are provided in Appendix A. Any disagreement was resolved by consensus between the two reviewers, and in cases where no consensus was achieved, another review author (TH) acted as an arbitrator. We assessed the risk of bias for the following domains: randomization process, deviations from intended interventions, missing outcome data, measurement of outcomes, selection of reported results. Assessors rated the risk of bias low, unclear, or high for all domains and overall [27].

In this study, we compared general physical activity promoting interventions with usual care. A meta-analysis was conducted for outcomes reported in two studies or more using a random-effects model, with a *p*-value of <0.05 considered as statistically significant. We calculated standardized outcomes (e.g., percentage time per day) if two studies reported outcomes of the same construct that were expressed in different units of measurement (e.g., minutes per day and percentage per day) or with different measurement instruments (e.g., performance tests and questionnaires). Standardized mean differences and weighted mean differences were calculated as part to compare outcomes in the meta-analysis. The statistical heterogeneity of the treatment effect among studies was assessed using the inconsistency I2 test, in which values greater than 50% were considered indicative of high heterogeneity [19]. Fisher’s exact test was used to calculate the relative effect of physical activity promoting interventions on reported falls. Analyses were performed using Stata software, version 15.0 (Stata Inc., College Station, TX, USA).

To assess the certainty of evidence, the Grading of Recommendations Assessment, Development, and Evaluation (GRADE) approach was used to rate the certainty of evidence for each outcome [28,29]. The GRADE-methodology is constructed upon five different items (study limitations; inconsistency of results; indirectness of evidence; imprecision of outcome estimates, and publication bias), which were independently assessed by two reviewers (JS and NK). Possible outcomes for each measure ranged from ‘very low certainty of evidence’ (we are very uncertain about the estimate), to ‘high certainty of evidence’ (further research is unlikely to change our confidence in the estimate of the effect) [28]. Any discrepancy in judgment was solved by consensus between the two reviewers, and in cases where no consensus was achieved, another review author (TH) acted as an arbitrator. Using the GRADEpro Guideline Development Tool (GRADEpro Guideline Development Tool (software), McMaster University, 2015, developed by Evidence Prime Inc., Hamilton, ON, Canada), a ‘summary of findings table’ was generated for all outcomes.

## 3. Results

Figure 1 shows a flow chart of the included studies.

Study characteristics of the four included trials are presented in Table 1. Details on the interventions are provided in Appendix A.

Brown et al. [30] included 100 patients (*n* = 8 loss to follow-up), aged 65 years or older who had a medical reason for admission. The mean age of the study population was 74 years (standard deviation (SD): 7 years) and included three females (3%). Common diagnoses included pneumonia, heart failure, and chronic obstructive pulmonary disease exacerbations. A standardized ambulation protocol was used for the intervention group, in which patients were assisted with ambulation up to twice daily. Besides, a behavioral strategy was used to encourage ambulation of patients by focusing on goal setting and addressing mobility barriers. Patients reported falls with a daily 24 h diary during the hospital stay. Activities of daily living were examined with a questionnaire on independent activities of daily living performance (ADL-activity) from 7 (independent) to 21 (total assistance). The community mobility was assessed with the Life-Space Assessment (0–120, higher scores representing greater mobility). Data on the time spent out of bed were not collected as a result of a technical failure of the accelerometry (brand: not reported).

Dall et al. [31] included 141 patients with a pulmonary diagnosis (*n* = 48 loss to follow-up). The mean age of the study participants was 73 years (SD: 13 years). The intervention consisted of visual feedback about daily time spent lying in bed, sitting, standing, and walking with a tablet. Methods for data collection on reported falls were not provided. Data on the time spent lying in bed, sitting, standing, and walking were collected in minutes per day with a tri-axial accelerometer (brand: not reported) supported with medical Band-aids.

Killey and Watt [32] included 77 patients with a minimum age of 70 years with provisional diagnoses including heart-, lung-, and diabetes-related morbidities (*n* = 29 loss to follow-up). The mean age of the study population was 83 years (SD: 7 years). The intervention was extra walking twice-daily, patients were instructed to walk the maximum distance they were able to comfortably cover. Methods for data collection on reported falls were not provided. ADL-activity performance was measured with the Barthel Index (range: 0–100, 100 as highest possible independence). Data on hospital mobility were collected as maximum distance walked in meters.

McGowan et al. [33] included 50 patients over the age of 65 years with an acute medical diagnosis (*n* = 2 loss to follow-up). The study participants had a mean age of 85 years (SD: 7 years). The pedal exercises intervention consisted of ‘5 min of chair-based pedal exercises three times a day with no specified targets on number of revolutions’ on an Able 2-pedal exerciser. ADL-activity was scored with the Elderly Mobility Scale, a 20-point ordinal scale from 0 ‘full dependent in mobility’ to 20 ‘independent’. Data on the number of revolutions were collected with a build-in pedometer and data on the time spent on the pedal exerciser with an accelerometer (ActivPal^®,^ developed by PAL Technologies, Glasgow, United Kingdom).

Risk of bias-2 scores are presented in Table 2.

The overall risk of bias score was ‘high’ for three studies [31,32,33]; one study scored ‘some concerns’ [30]. The randomization process was judged to be of high risk of bias in two studies as a result of no random allocation sequence and no concealed allocation [31,32]. None of the studies provided information on deviations from intended interventions, resulting in some concerns in all included studies [30,31,32,33]. Two studies were judged to be of high risk of bias due to missing outcome data, because both studies had a high loss to follow-up and did not perform an intention-to-treat analysis or missing data analysis [32]. The study of Dall et al. [31] showed a high risk of bias regarding measurement of outcomes, as they provided no details on the methods for reported falls and no information on the psychometric quality of the accelerometry used. Selection of reported results raised some concerns in three studies, as these studies provided no study protocol [32], data on within-group differences [30], and data on between-group differences [33].

Table 3 shows a summary of findings for each outcome.

The reported falls were assessed in three studies [30,31,32]. Patients in the control groups reported 5 falls per 126 patients in contrast to 0 falls per 122 patients with the general physical activity promoting interventions, showing no statistically significant difference (*p*: 0.06). The certainty of evidence for reported falls was ‘moderate’ as a result of the indirectness of outcomes (reported falls are a surrogate outcome for actual falls incidents). The ADL-activity performance was examined in three studies, comprising data of 203 participants [30,32,33]. The standardized mean difference ADL-activity performance was −0.07 (95% CI: −0.64 to 0.51), showing no statistically significant difference between groups (Figure 2). The certainty of evidence for ADL-activity was ‘low’ due to study limitations (attrition of participants) and imprecision (large confidence intervals around the estimated mean).Two studies analyzed the time spent standing and walking during the hospital stay, comprising data of 141 participants [31,33]. The weighted mean difference time spent standing and walking was 2.0% per day (95% CI: −2.8% to 6.9%) showing no statistically significant difference between groups (Figure 3). The certainty of evidence for time spent standing and walking was ‘very low’ as a result of study limitations (unknown psychometric quality of accelerometry), imprecision (high standard deviations from the estimated mean), and inconsistent outcomes (results in two studies showed opposite standardized mean differences). None of the included studies reported hospital outcomes.

## 4. Discussion

In this systematic review and meta-analysis, we specifically studied the outcomes of general physical activity promoting interventions in patients hospitalized over 48 h. After an extensive literature search, we identified just four studies that met our eligibility criteria. We found no statistically significant effect (moderate certainty of evidence) of general physical activity promoting interventions on reported falls during the hospital stay. Besides, we found no statistically significant effect (very low certainty of evidence) of interventions on physical activity in patients enrolled in general physical activity promoting interventions compared to usual care. Finally, we found no statistically significant effect (low certainty of evidence) for physical activity promoting interventions on ADL-activity in patients during their hospital stay.

To our knowledge, this is the first systematic review of randomized controlled trials to study the effectiveness of interventions aiming to promote physical activity in the entire hospital population without additional supervision of specialized staff. We specifically looked at activity promoting interventions that could be employed in the whole hospital for all patients without additional specialized supervision, as the workload of hospital personnel is already perceived as high [34]. The most similar systematic review studied the effects of supervised activity interventions in older hospitalized patients [35]. They concluded, without a GRADE analysis, that evidence for the effect of physical interventions on physical performance in older patients during hospitalization was uncertain, which is in line with our findings. The use of general physical activity promoting interventions might be promising looking at the direction of reported falls; however, there is little evidence available. Hypothetically, this could mean that people admitted to the hospital do not necessarily need supervised general interventions; they have more to gain from interventions that focus on the context in which physical activity care is provided [36].

In our review and meta-analysis, we aimed to include hospitalized patients of all ages; however, the subjects included in our review are all of older age. This shift towards older patients is likely caused by the fact that we did not include disease-specific interventions, in which more patients of varying age are present. We hypothesize that this might have overestimated the findings of our review to some extent. After all, hospitalized older patients have an increased risk for developing hospital-associated disability [37], and ADL dependence compared to younger patients [38]. On the one hand, it is thought that promoting physical activity in this population with older patients has the potential to elicit greater effects, as the consequences of physical activity are more pronounced. On the other hand, the physical inactivity epidemic targets all patients in the hospital and does not discriminate for age [1]. Although frail older patients are more prone to the negative consequences of inactivity during hospitalization than the relatively younger patients, there is evidence suggesting that almost half of the relatively younger patients were significantly affected by sedentary behavior during hospitalization [14]. A recent systematic review and meta-analysis by Fazio et al. [39] confirms that a broad spectrum of inpatient populations is physically inactive with a point estimate of 70 min walking a day (interquartile range: 58–83 min). Physical activity levels of patients during their hospital stay are low and do not seem to be associated with their level of illness [2,40,41]; however, it seems reasonable that the level of illness is somehow related to patients’ ability and willingness to be physically active [42,43]. In other words, interventions targeting the entire hospital population might have a greater impact as a whole.

There are several explanations for the limited effectiveness of the interventions in terms of reported falls, ADL-activity, and time spent standing and walking. First, we included few studies that might compromise the validity of meta-analysis [44]. Second, interventions were not supervised, which means that activities and outcomes might be under- or overestimated. Third, the practice of any physical activity requires a minimum of time and exposure to take advantage of it [11]. It is not known how often the patients in the included studies exceeded a minimum of time. A fourth explanation is that the interventions might have been too simplistic. All included interventions have mono-faceted treatment approaches [30,32], while the underlying mechanism that triggers physically inactive behavior in the hospital is multi-faceted. Previous research demonstrates that physical inactivity is more than just patient-related characteristics such as functional status, pain, and shortness of breath [9,45]. Inactivity can be triggered by the built environment of the hospital (e.g., the inactivating hospital bed centric approach to care, the lack of privacy and shelter in a hospital room), the (lack of) materials to mobilize patients (e.g., being connected to drains or catheters, lack of chairs to sit, lack of rollers to walk), and the mindset of both patients (e.g., beliefs that to be in the hospital is to be in bed or that patients are not welcome outside of their rooms) and healthcare professionals (e.g., dedication to engage patients in physical activity or thoughts that patients are better off lying in bed) [46,47,48]. Given this plethora of related variables, one could expect a multi-faceted approach to tackle physical inactivity in the hospital setting. Future researchers might need to view physical activity promoting interventions in the hospital as complex interventions, which need to be developed and evaluated accordingly [49,50].

Our systematic review and meta-analysis have an important limitation that needs to be addressed, namely, that only four studies were eligible for inclusion. One of the reasons was the inclusion of randomized controlled trials only. We have excluded 10 full-text studies based on study design, because these studies did not have a randomized controlled trial design to study intervention effect estimates. However, the excluded studies may contain useful information and important outcomes related to general physical activity promoting interventions. Nevertheless, to our knowledge, even if we had broadened our inclusion criteria in terms of the study design with, for example, before–after studies, no additional studies in hospitalized patients have been identified.

Even though the effects of physical activity promoting interventions remain largely uncertain, healthcare professionals should still take the detrimental consequences of physical inactivity during the hospital stay seriously [7,8,9,51]. The moderate certainty of evidence that these general, unsupervised interventions reduce the number of falls, should promote healthcare professionals to explore the possibilities of promoting physical activity. It is important that healthcare professionals appreciated the complex nature of physical inactivity in the hospital and understand that this behavior is ingrained in both patients’ expectations as well as the build hospital environment [46,52]. Researchers might consider the use of cluster randomized controlled trial designs or pragmatic, quasi-interrupted time series to further study the effect of general physical activity promoting interventions in the complex hospital context [53].

## 5. Conclusions

In conclusion, we found no statistically significant effect of general physical activity promoting interventions on functional outcomes. The meta-analysis showed no statistically significant difference of reported falls between participants in the intervention and control groups with a moderate certainty of evidence. No statistically significant difference between intervention and control groups was found for ADL-activity and time spent standing and walking with a (very) low certainty of evidence. Although general physical activity promoting interventions might have positive effects on functional outcomes in patients hospitalized over 48 h according to observational studies, currently, there is a lack of well-designed experimental studies to make recommendations with a high degree of certainty of evidence.

## Figures and Tables

**Figure 1 ijerph-18-01233-f001:**
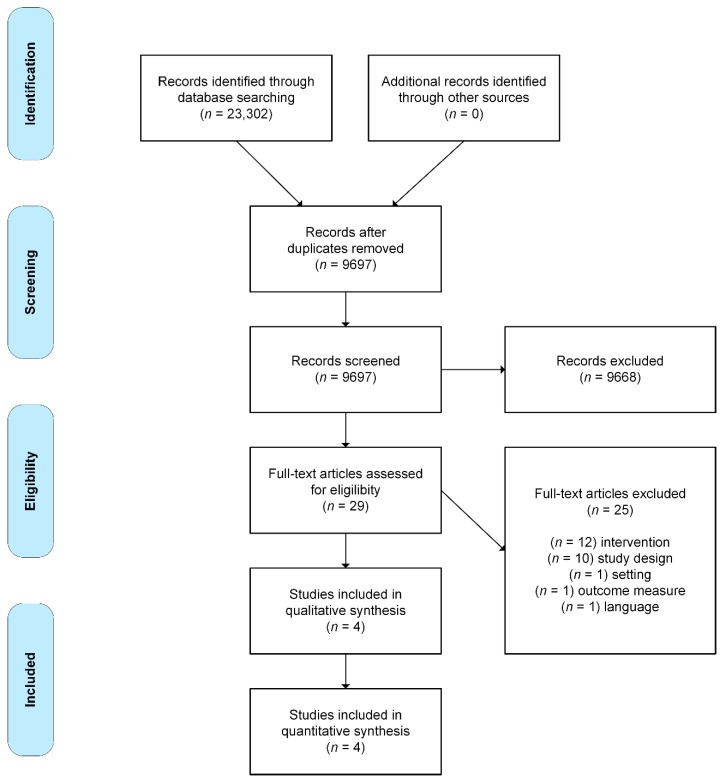
Flow chart of the study selection process, adapted from Moher et al. [18].

**Figure 2 ijerph-18-01233-f002:**
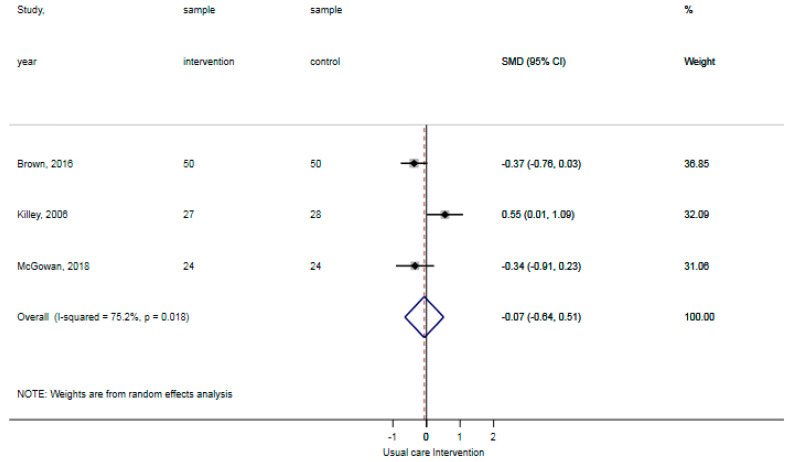
Forst plot activities of daily living (ADL)-activity performance showing no statistically significant difference between usual care and intervention.

**Figure 3 ijerph-18-01233-f003:**
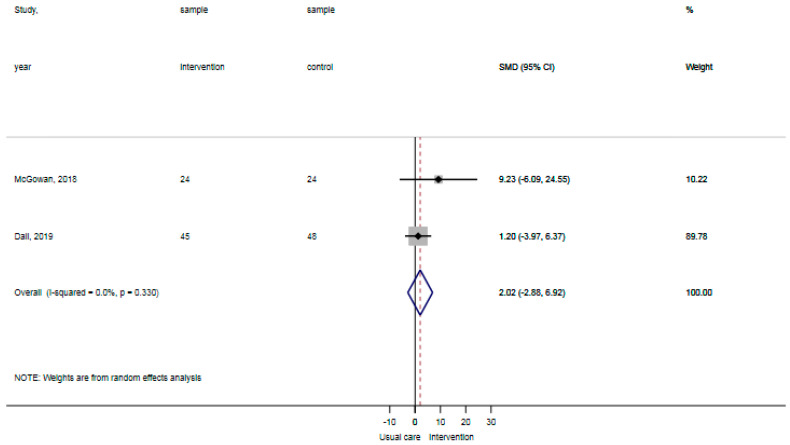
Forst plot time spent standing and walking showing no statistically significant difference between usual care and intervention.

**Table 1 ijerph-18-01233-t001:** Study characteristics, population, intervention versus control, and outcomes.

First Author (Year)	Country, Setting	Study Population	Length of Hospital Stay in Days	Intervention Versus Control	Outcomes and Measures	Results
Brown et al. (2016) [30]	USA, medical ward	Patients were 65 years or older, admitted with a medical diagnosis (*n* = 100 at baseline, *n* = 92 at follow-up)	UC: 3.6 (SD: 2.4) MP: 4.6 (SD: 4.0)	Twice-daily ambulation and a behavioral strategy versus twice-daily visits (usual care)	Reported falls during the hospital stay with 24-h diary (number)	UC = 3 falls; MP = 0 falls
					ADL-activity performance with Functional Outcome Assessment (score)	UC: t0 = 8.7 (SD: 0.3); UC: t1 = 8.0 (SD: 0.3); MP: t0 = 8.4 (SD: 0.3); MP: t1 = 8.1 (SD: 0.3)
					Community mobility with Life-Space Assessment tool (score)	UC: t0 = 51.5 (SD: 3.0); UC: t2 = 41.8 (SD: 3.2); MP: t0 = 54.0 (SD: 4.2); MP: t2 = 52.6 (SD: 4.4)
					Time spent out of bed using accelerometer (minutes per day)	No data due to technical failure
Dall et al. (2019) [31]	Denmark, pulmonary ward	Patients were admitted with a pulmonary diagnosis (*n* = 141 at baseline, *n* = 93 at follow-up)	UC: 8.3 (SD: 10.4) VF: 7.3 (SD: 12.2)	Visual feedback of the daily time spent in bed, sitting, standing, and walking; versus no feedback	Reported falls during the hospital stay (number)	UC: 0 falls; VF: 0 falls
					Time spent standing and walking using accelerometers (minutes per day)	UC: 64 (−3 to 131); VF: 81 (46 to 117)
					Time spent lying in bed and sitting using accelerometer (minutes per day)	UC: 1376 (95% CI: 1309 to 1443); VF: 1359 (95% CI: 1323 to 1394)
Killey and Watt (2006) [32]	Australia, medical ward	Patients were 70 years or older, admitted with provisional diagnoses (*n* = 77 at baseline, *n* = 48 at follow-up)	Not reported	Twice a day seven days a week extra walking to comfortable limit versus standard assistance to walk (normal care)	Reported falls during the hospital stay (number)	UC: 2 falls; MP: 0 falls
					ADL-activity performance with Barthel index (score)	UC: t0 = 58.1 (SD: 27.8); UC: t1 = 55.2 (SD: 31.8); MP: t0 = 59.2 (SD: 25.9); MP: t1 = 70.8 (SD: 24.3)
					Hospital mobility with performed maximum distance walked (meters)	UC: t0 = 32.1 (SD: 32.8); UC: t1 = 47.9 (SD: 47.7); MP: t0 = 38.6 (SD: 27.1); MP: t1 = 79.4 (SD: 58.0)
McGowan et al. (2018) [33]	UK, acute medical ward	Patients were 65 years or older, admitted with an acute medical diagnosis (*n* = 50 at baseline, *n* = 48 at follow-up)	Not reported	Pedal exercises for 5 min three times a day with minimal supervision versus usual level of clinical input (standard care)	ADL-activity performance with Elderly Mobility Scale (score)	UC: t0 = 15.7 (SD: 4.0) UC: t1 = 14.1 (SD: 2.9) PE: t0 = 13.8 (SD: 4.6) PE: t1 = 13.2 (SD: 2.8)
					Time spent standing and walking using accelerometer (percentage per day)	UC: 5.0% (IQR: 0.6% to 17.1%); PE: 4.5% (IQR: 0.1% to 45.8%)
					Revolutions cycled using Able-2 exerciser (number)	PE: 152 (IQR: 43.5 to 464.5)
					Pedal exercises using Able-2 exerciser (minutes)	PE: 5.08 (95% CI: 2.03 to 20.05)

ADL: activities of daily living; CI: confidence interval; IQR: interquartile range; MP: mobility program; *n*: number; PE: pedal exercises; SD: standard deviation; t0: baseline; t1: hospital discharge; UC: usual care; UK: United Kingdom; USA, United States of America; VF: visual feedback.

**Table 2 ijerph-18-01233-t002:** Risk of bias-2 scores of the included studies.

First Author	Randomization Process	Deviations from Intended Interventions	Missing Outcome Data	Measurement of Outcomes	Selection of Reported Results	Overall
Brown et al. [30]	+	?	+	+	?	?
Dall et al. [31]	–	?	+	–	+	–
Killey and Watt [32]	–	?	–	+	?	–
McGowan et al. [33]	+	?	–	+	?	–

+, low risk; ?, some concerns; – high risk.

**Table 3 ijerph-18-01233-t003:** Summary of findings according to Grading of Recommendations Assessment, Development, and Evaluation (GRADE) certainty of evidence.

A General Physical Activity Promoting Intervention Compared to Usual Care in Patients During the Hospital Stay					
Patient or Population: Patients During the Hospital Stay; Setting: Hospital Care; Intervention: A General Physical Activity Promoting Intervention; Comparison: Usual Care					
	Anticipated Absolute Effects * (95% CI)				
Outcomes	Risk with Usual Care	Risk with a Mobility Program	Number of Participants (Studies)	Certainty of Evidence (GRADE)	Comments
Reported falls	5 per 126	The number of reported falls in the intervention group was 0 per 122 (0 to 0), *p* = 0.06 **	248 (3 RCTs)	⨁⨁⨁◯ MODERATE ^a^	A general physical activity promoting intervention probably results in a slight reduction in reported falls.
ADL-activity performance assessed with: Functional Outcome Assessment, Barthel Index or Elderly Mobility Scale	The standardized mean ADL-activity performance was 8.0 (7.4 to 8.6)	The standardized mean difference ADL-activity performance in the intervention group was 0.07 lower (−0.64 to 0.51)	203 (3 RCTs)	⨁⨁◯◯ LOW ^b,c^	The evidence is uncertain about the effect of a general physical activity promoting intervention on ADL-activity.
Time spent standing and walking assessed with accelerometry	The weighted mean percentage time spent standing and walking was 4.8% (0.4 to 28.3%)	The weighted mean percentage time spent standing and walking in the intervention group was 2.0% higher (−2.8% to 6.9%)	141 (2 RCTs)	⨁◯◯◯ VERY LOW ^b,c,d^	The evidence is very uncertain about the effect of a general physical activity promoting intervention on time spent standing and walking.

* The risk in the intervention group (and its 95% confidence interval) is based on the assumed risk in the comparison group and the relative effect of the intervention (and its 95% CI). ** Calculated with Fisher’s exact test. Abbreviations, ADL: activities of daily living; CI: Confidence interval; IQR: interquartile range, RCTs: randomized controlled trials. Explanations: ^a^ The indirectness was probable as the outcome measures (reported falls) are surrogate outcomes for actual fall incidents; ^b^ The risk of bias was serious, as reflected by important study limitations such as the unknown psychometric quality of the measurement instruments, attrition of participants, and unknown deviations from interventions (see Risk of Bias-2 assessment); ^c^ The imprecision was serious, as reflected by high standard deviations around the estimated mean; ^d^ The inconsistency was serious as some results show opposite standardized mean differences.

## Data Availability

The data that support the findings of this study are available from the corresponding author, N.K., upon reasonable request.

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
