# Peer review of "Effects of General Physical Activity Promoting Interventions on Functional Outcomes in Patients Hospitalized over 48 Hours: A Systematic Review and Meta-Analysis of Randomized Controlled Trials"

_ijerph, 2021, doi:10.3390/ijerph18031233_

Round 1

Reviewer 1 Report

I am happy to review this paper again and the authors have addressed my concerns. but I still suggest makeing the format of paper better according to the journal's requirements. 

Author Response

Reviewer 1 recommendation:

  1. I am happy to review this paper again and the authors have addressed my concerns. but I still suggest makeing the format of paper better according to the journal's requirements.

Authors’ response:

We are happy to hear that we have resolved all concerns. We notice that the format of our paper has already been improved by the editorial office. We rechecked the authors guidelines and revised our manuscript to fit the journal's requirements.

Reviewer 2 Report

Firstly, I would like to congratulate the authors for the rewriting and corrections made to the manuscript. The increase in quality is noticeable and admirable. However, there are still some points that need attention.

- The presentation of the results of the meta-analysis is superficial and incomplete. Although the option of presenting the results is the choice of the authors, I believe that the presentation of meta-analyzes would be better with the use of the forest plot. In the forest plot, it is possible to visualize important information that isn't present in the results, such as the confidence intervals of each study included in the meta-analysis, which can help us understand the reason for the almost statistical significance and high heterogeneity. Also, it is possible to analyze the weight of each included study and compare it with the risk of bias and the certainty of the evidence (GRADE analysis). For these and other factors, I suggest using the forest plot.
However, if the author doesn't want to comply with this suggestion, I ask you to at least add the p-value, the confidence interval, and the heterogeneity of the three meta-analyses, and not only in one or the other, as it is in the current version of the manuscript.

- As the practice of any physical activity requires a minimum amount of time and exposure for there to be a benefit, I suggest that in addition to including the average length of stay for patients in table 1, and this should be taken into consideration in the discussion.

- As the exercises were not supervised, and the results can be under or overestimated (it can also be mentioned as a limitation of the study). This, together with a probable low hospital stay, can also be taken into account during the discussion, to explain the almost statistical significance.

Author Response

Reviewer 2 recommendation:

  1. The presentation of the results of the meta-analysis is superficial and incomplete. Although the option of presenting the results is the choice of the authors, I believe that the presentation of meta-analyzes would be better with the use of the forest plot. In the forest plot, it is possible to visualize important information that isn't present in the results, such as the confidence intervals of each study included in the meta-analysis, which can help us understand the reason for the almost statistical significance and high heterogeneity. Also, it is possible to analyze the weight of each included study and compare it with the risk of bias and the certainty of the evidence (GRADE analysis). For these and other factors, I suggest using the forest plot.

Authors’ response:

We would like to thank Reviewer 2 for the recommendations and feedback. We have revised our manuscript and added forest plots to visualize important information that was not present in the results. The forest plots are now presented as Figures 2 and 3. We have weighted each outcome and related it to the certainty of the evidence in accordance with the GRADE methodology in Table 3.

Reviewer 2 recommendation:

  1. As the practice of any physical activity requires a minimum amount of time and exposure for there to be a benefit, I suggest that in addition to including the average length of stay for patients in table 1, and this should be taken into consideration in the discussion.

Authors’ response:

We agree that any physical activity required a minimum amount of time and exposure to benefit, as stated in the New Physical Activity Guidelines by Thompson and Eijsvogels. We decide not to state a number, because this minimum amount of time is not yet known for the hospital population. We do thank the author for the suggestion and added:

  • Discussion, lines 290-292: Third, the practice of any physical activity requires a minimum of time and exposure to take advantage of it.[11] It is not known how often the patients in the included studies exceeded a minimum of time.
  • Table 1. Length of hospital stay in days.

Reviewer 2 recommendation:

  1. As the exercises were not supervised, and the results can be under or overestimated (it can also be mentioned as a limitation of the study). This, together with a probable low hospital stay, can also be taken into account during the discussion, to explain the almost statistical significance.

Authors’ response:

The included general physical activity promoting interventions were not supervised and, therefore, activities and outcomes may be under- or overestimated. We agree that this is an important limitation to present. We added:

  • Discussion, lines 287-289: Second, interventions were not supervised which means that activities and outcomes might be under- or overestimated.

It is difficult to hypothesize a relationship between (a probable) low hospital stay and functional outcomes, as length of hospital stay was not reported in two studies. Furthermore, based on our results, we cannot conclude that supervision or a longer/shorter hospital stay of patients could lead to statistical significant differences on functional outcomes between usual care and intervention.

Reviewer 3 Report

While the authors have addressed the comments, the fundamental issue cannot be addressed, that is the premature state of this review. The conclusions are based on very few studies in which there is huge variability in the outcomes hence the conclusions might be unreliable.

Author Response

Reviewer 3 recommendation:

  1. While the authors have addressed the comments, the fundamental issue cannot be addressed, that is the premature state of this review. The conclusions are based on very few studies in which there is huge variability in the outcomes hence the conclusions might be unreliable.

Authors’ response:

As stated in our appeal letter, we disagree with this opinion since a properly conducted systematic review will inform clinicians, policy makers and other researchers on the current state of the evidence. This also applies to situations where only a limited number of randomized controlled trials is available. The unreliability or uncertainty is considered and taken care of in the evaluation of the evidence according to the GRADE methodology. It should be noted that even in systematic reviews with a large number of primary studies the evidence can be uncertain (unreliable) when methodological shortcomings are prevalent.

Round 2

Reviewer 3 Report

As stated originally, this review is premature and the conclusions cannot be warranted based on 4 studies.

Author Response

This manuscript is a resubmission of an earlier submission. The following is a list of the peer review reports and author responses from that submission.

Round 1

Reviewer 1 Report

see the comments from the attached

Reviewer 2 Report

First, I would like to congratulate the authors for their persistence in conducting this work, I realized that the protocol was registered on PROSPERO in 2017, and underwent improvements in 2018 and 2019. Now let's get to what matters to us. At first, I found the idea of ​​the article very interesting, however, when I arrived at the methodology and compared some points with the PROSPERO register and the current recommendations for systematic review and meta-analysis, I was not satisfied. That is, I believe that the article has much to improve. Specific comments can be seen below.

Specific comments:

  • Title. I suggest: "Effects of General Physical Activity Promoting Interventions on Functional and hospital Outcomes in Patients hospitalized over 48 hours: A Systematic Review with Meta-Analysis", or "Effects of General Physical Activity Promoting Interventions on Functional and hospital Outcomes in older adults hospitalized over 48 hours: A Systematic Review with Meta-Analysis"
  • Line 49. As it is the objective of the work, the functional outcomes that will be analyzed must be included, or leave only "functional outcomes" and describe them in the methods.
  • Line 50. As it is the objective of the work, the hospital outcomes that will be analyzed must be included, or leave only "hospital outcomes" and describe them in the methods.
  • Line 51. Instead of inserting "hospitalized adults of all ages", I suggest one of two options: "patients hospitalized over 48 hours" (according to PROSPERO's record); or "older adults hospitalized over 48 hours" (since the 4 studies included are in the older adults).
  • Line 54. The Cochrane handbook version is out of date. The latest online version is the 6.0, updated in July 2019. Available from www.training.cochrane.org/handbook.

Online Handbook:
Higgins JPT, Thomas J, Chandler J, Cumpston M, Li T, Page MJ, Welch VA (editors). Cochrane Handbook for Systematic Reviews of Interventions version 6.0 (updated July 2019). Cochrane, 2019. Available from www.training.cochrane.org/handbook.
Print edition:
Higgins JPT, Thomas J, Chandler J, Cumpston M, Li T, Page MJ, Welch VA (editors). Cochrane Handbook for Systematic Reviews of Interventions. 2nd Edition. Chichester (UK): John Wiley & Sons, 2019.

  • Line 63. Published articles between inception and January 2020? Inception of? Please correct this sentence and include the complete information.
  • Lines 65 to 70. Instead of putting examples of terms of the population, intervention, and outcomes, I recommend describing population, intervention, and outcomes as Voet et. al. did on pages 24 and 25 (https://doi.org/10.1002/14651858.CD003907.pub5), maintaining the supplementary material as it is; or else, I suggest adding the table with the terms used, which is in the supplementary material, in the main document of the study.
  • Line 78. Indicate only the two who did the screening.
  • Line 81. Same authors of screening peer titles and abstracts?
  • Line 89. Remove "for example" and put what was used.
  • Line 99. The tool for analyzing the risk of bias was also updated last year. Version 2 of the Cochrane risk-of-bias tool for randomized trials (RoB 2), is described in chapter 8 of the Cochrane Handbook (Higgins JPT, Savović J, Page MJ, Elbers RG, Sterne JAC, should now be used. : Assessing the risk of bias in a randomized trial. In: Higgins JPT, Thomas J, Chandler J, Cumpston M, Li T, Page MJ, Welch VA (editors). Cochrane Handbook for Systematic Reviews of Interventions version 6.0 (updated July 2019). Cochrane, 2019. Available from https://training.cochrane.org/handbook/current/chapter-08), where the link to RoB2 can be found (https://www.riskofbias.info/welcome/rob- 2-0-tool). Thus, the analysis of the risk of bias should be redone.
  • Lines 104 to 107. If are still going to use these analyzes, redo to be consistent with RoB2.
  • Lines 134 to 142. Repeated information in the text and flowchart. I suggest removing these lines and leaving only the flowchart.
  • Line 179 (Table 1). This table does not provide sufficient characteristics of the study, population, and outcomes. The paragraphs preceding the table are much more complete, please complete the table with the information described in lines 92 to 95.
  • Lines 183 to 189. Redo according to RoB2.
  • Line 190 (Table 2). Redo according to RoB2.
  • Lines 193 to 227. The information in the text is not sufficient to understand all the statistical analyzes conducted (meta-analyzes). My suggestion is to insert the forest plots (meta-analysis graph), however, if the authors are not interested in doing this, they should at least insert the necessary information for the minimum understanding of the analyzes (number of studies, number of participants in each group (sum the number of each study), p-value, 95% CI, I2 test, whether the analysis was performed using the MD or SMD, with random or fixed effect).
  • Lines 208 to 227. "Meta-analysis is the statistical combination of results from two or more separate studies." (Deeks JJ, Higgins JPT, Altman DG (editors). Chapter 10: Analyzing data and undertaking meta-analyses. In: Higgins JPT, Thomas J, Chandler J, Cumpston M, Li T, Page MJ, Welch VA (editors). Cochrane Handbook for Systematic Reviews of Interventions version 6.0 (updated July 2019). Cochrane, 2019. Available from https://training.cochrane.org/handbook/current/chapter-10). Therefore, the outcomes evaluated with only one study should be removed from the quantitative synthesis (meta-analysis), remaining only in the qualitative synthesis.
  • Line 228 (Table 3). After the previous adjustments, redo to confirm the results.
  • Line 304. In conclusion, don't just mention falls. Talk about all the outcomes, or make it more general (including all).
  • Line 308 and 309. I suggest making this recommendation in the discussion, along with the study limitations.
  • After correcting the points above, I suggest reworking the summary and analyzing possible changes in the discussion, so that both are compatible with the changes made on methods, results, and conclusion.

Reviewer 3 Report

It could be more refined and defined in terms of age and comorbidity characteristics. Type of exercise could be ellaborated. 

Functional characteristics could be defined. 

Round 2

Reviewer 2 Report

The results paragraph of the meta-analyzes does not provide enough data to fully visualize and understand the results. I suggest that the authors rethink the strategy of the absence of forest plots.

Reviewer 4 Report

I understand the authors' reply but all in all I am not convinced of the timeliness, scope, and validity of the review. I upheld my original opinion and evaluation vis a vis the authors' responses to the criticisms.